# Monitoring Brain State and Behavioral Performance during Repetitive Visual Stimulation

**Alexander K. Kuc** [1,*,†], **Semen A. Kurkin** [1,2,†], **Vladimir A. Maksimenko** [1,2,3,4,†], **Alexander N. Pisarchik** [2,5]
and **Alexander E. Hramov** [1,2,3,4]

1  Baltic Center for Artificial Intelligence and Neurotechnology, Immanuel Kant Baltic Federal University, 236016 Kaliningrad, Russia; s.kurkin@innopolis.ru (S.A.K.); v.maksimenko@innopolis.ru (V.A.M.); a.hramov@innopolis.ru (A.E.H.)
2  Center for Technologies in Robotics and Mechatronics Components, Innopolis University, 420500 Innopolis, Russia; alexander.pisarchik@ctb.upm.es
3  Neurotechnology Deparment, Lobachevsky State University of Nizhny Novgorod, 603022 Nizhny Novgorod, Russia
4  Department of Innovative Cardiological Information Technology, Institute of Cardiological Research, Saratov State Medical University, 410012 Saratov, Russia
5  Centre for Biomedical Technology, Universidad Piolitécnica de Madrid, Pozuelo de Alarcón, 28223 Madrid, Spain
*  Correspondence: plo@sstu.ru
†  These authors contributed equally to this work.

**Abstract:** We tested whether changes in prestimulus neural activity predict behavioral performance (decision time and errors) during a prolonged visual task. The task was to classify ambiguous stimuli—Necker cubes; manipulating the degree of ambiguity from low ambiguity (LA) to high ambiguity (HA) changed the task difficulty. First, we assumed that the observer's state changes over time, which leads to a change in the prestimulus brain activity. Second, we supposed that the prestimulus state produces a different effect on behavioral performance depending on the task demands. Monitoring behavioral responses, we revealed that the observer's decision time decreased for both LA and HA stimuli during the task performance. The number of perceptual errors lowered for HA, but not for LA stimuli. EEG analysis revealed an increase in the prestimulus 9–11 Hz EEG power with task time. Finally, we found associations between the behavioral and neural estimates. The prestimulus EEG power negatively correlated with the decision time for LA stimuli and the erroneous responses rate for HA stimuli. The obtained results confirm that monitoring prestimulus EEG power enables predicting perceptual performance on the behavioral level. The observed different time-on-task effects on the LA and HA stimuli processing may shed light on the features of ambiguous perception.

**Keywords:** ambiguous stimuli; Necker cubes; classification task; EEG analysis; wavelet analysis; decision time; perceptual errors; time-on-task effect



## 1. Introduction

Sensory processing is a fundamental brain function that allows us to more easily interact with each other and with our environment. In everyday life, we collect sensory data and process it for interpretation and decision making [1]. The accuracy and timeliness of our decisions depend on the speed and correctness of sensory processing. The effectiveness of sensory processing, in turn, is determined by a number of exogenous and endogenous factors [2]. In particular, the exogenous component reflects the quality of the sensory input. Thus, when faced with unambiguous information, we can easily interpret it. On the contrary, when information becomes ambiguous, interpreting it takes more effort.

In turn, the endogenous component depends on the state of the person; on their attention, fatigue, and subjective experience [3]. In many experimental studies where ambiguous stimuli were used, endogenous effects were found to be especially pronounced

when the sensory information quality was low [4]. Therefore, the observer must concentrate to gather more information to make the right decision, relying on personal experience to extrapolate limited information or unresolve its ambiguity.

The conditions under which the observer receives and processes information are important as well. For example, high-speed driving on a rainy night requires the very fast processing of low-quality information. Performing monotonous tasks with increased responsibility (e.g., flight or power plant operators) also requires maintaining high performance and emergence preparedness. In these stressful conditions, the influence of exogenous and endogenous factors on the likelihood of perceptual errors should be considered. Therefore, knowing and monitoring these factors can help predict perceptual errors and reduce their probability. Furthermore, human condition monitoring (endogenous factor) is a task for passive brain–computer interfaces (BCIs) [5]. Unlike the traditional active BCI, which issues control commands through mental intent, passive BCIs continuously monitor the brain state during extended periods of cognitive activity and signal if it deviates from the normal state [6].

To control the quality of received information and its processing, the BCI must track both exogenous and endogenous components. Thus, the BCI must meet the task requirements (exogenous factor) along with the neural activity (endogenous factor). Moreover, objective assessments of the task requirements may depend on the amount of information, its ambiguity, and multimodality. Subjective estimates can be derived from the observer's reaction, such as response time, eye movements, and other behavioral indicators [7]. To take into account all these processes, it is necessary to move from a passive to a reactive BCI. The latter uses stimuli and analyzes the brain state through time intervals assigned to them [8]. Following this concept, a reactive BCI should analyze the brain state while performing a task. This will provide information on the influence of endogenous and exogenous factors on the speed and quality of information processing.

The further development of BCIs aims not only at the detection, but also the prediction of the human states. These BCIs will give rise to the artificial intelligence systems that assist or alarm when detecting a high probability of critical errors. Developing such systems requires finding the associations between the current state of the BCI operator and their performance in solving ongoing tasks. A bulk of literature associates changes in the human condition with their behavioral performance in ongoing tasks. In particular, attention, a fundamental aspect of the observer's state, modulates prestimulus alpha- and beta-band power [9–11], influencing the accuracy of perceptual decisions. Thus, either medium or low alpha- and high beta-band power during the prestimulus period is beneficial for sensory perception [11,12]. According to [13], the power and the prestimulus EEG phase coupling in the alpha- and beta-bands affect visual perception performance. Namely, better performance is associated with low phase coupling in the alpha-band and high phase coupling in the beta-band. Recent work [14] revealed that EEG power in the beta-2 frequency band at rest negatively correlated with the response times in the ongoing attentional task. While most works reported correlations between the neural correlates averaged across the trials, or between event-related potentials, in recent work [15], the authors used EEG power in different bands to predict individual performance in single trials contributing to the BCI problem.

Complementing the existing literature, we examined how the prestimulus EEG power predicts behavioral performance depending on the task demands. We considered a long-lasting monotonous experiment in which the participant perceived ambiguous stimuli and reported on each stimulus interpretation with the joystick buttons. The visual stimulus was an ambiguous Necker cube. The inner edges contrast defines one of two possible cube's orientations, left or right, and determines stimulus ambiguity. When ambiguity is low, cubes morphology is different for the left and right orientations. Therefore, subjects easily report the correct one. For the high ambiguity, stimulus morphology becomes similar for different orientations; therefore, subjects spent more effect to find the differences. In the recent works, we observed that subjects responded faster to the Necker cubes presented

at the end of the experiment [16]. We also found that the brain utilized different neural mechanisms when processing stimuli with low and high ambiguity [2,4,17]. Based on these results, we hypothesized that during a long experiment with the Necker cubes, the observer's state changed, causing changes in behavioral performance. We expected to find the neural correlates of these changes in the prestimulus state and use them to predict the performance of the ongoing stimulus. We also supposed that changes in the human condition had a different effect depending on the stimulus ambiguity.

To test this hypothesis, we tracked the behavioral characteristics (decision times and errors) and simultaneously detected the EEG signals during a long monotonous task, including the Necker cubes interpretations. Behavioral monitoring revealed that decision time decreased with time on task, despite the ambiguity. For high ambiguity, we also observed a reduction of perceptual errors. EEG analysis showed growing prestimulus 9–11 Hz EEG power in the right temporal region. This EEG power negatively correlated with the decision time to the stimuli, with low ambiguity and the erroneous responses rate to stimuli with high ambiguity. The obtained results confirm that monitoring prestimulus EEG power enables predicting perceptual performance on the behavioral level.

## 2. Materials and Methods

### 2.1. Participants

Twenty healthy volunteers (nine females, 26–35 y.o.) with normal or corrected-to-normal vision participated in the experiments after providing written informed consent. Participants took part in similar experiments not earlier than six months before. All experiments were carried out in accordance with the requirements of the Declaration of Helsinki and approved by the local Research Ethics Committee of the Innopolis University.

### 2.2. Visual Stimuli

We used an experimental paradigm with an ambiguous bistable visual stimulus in the form of the Necker cube, which allows two possible interpretations [4,16,18,19]. The non-perceptually impaired volunteer interpreted this two-dimensional (2D) image as a three-dimensional (3D) object which is oriented either left or right. The balance between the brightness of three inner lines (1,2,3) located in the left bottom corner and three inner lines (4,5,6) in the right upper corner determines the ambiguity and orientation of the 3D cube (Figure 1A). The contrast parameter $a \in [0,1]$ is the normalized brightness of the inner lines (1,2,3) in the grayscale palette. In turn, the normalized brightness of the other inner lines (4,5,6) is defined as $1 - a$. Thus, the limiting cases $a = 0$ and $a = 1$ correspond to unambiguous 2D projections of the cube oriented to the left or to the right, respectively, whereas $a = 0.5$ implies a completely ambiguous spatial orientation of the 3D cube.

In the experiment, we used a set of the Necker cube images with $a = \{0.15, 0.25, 0.4, 0.45, 0.55, 0.6, 0.75, 0.85\}$ (Figure 1B), which we divided by subsets of cubes oriented to the left $a = \{0.15, 0.25, 0.4, 0.45\}$ and to the right $a = \{0.55, 0.6, 0.75, 0.85\}$. At this set, stimuli with low ambiguity (LA) $a = \{0.15, 0.25, 0.75, 0.85\}$, are easily interpreted, whereas the interpretation of stimuli with high ambiguity (HA) $a = \{0.40, 0.45, 0.55, 0.60\}$ requires more effort [4]. We also assume that HA processing engages more top-down control [20].

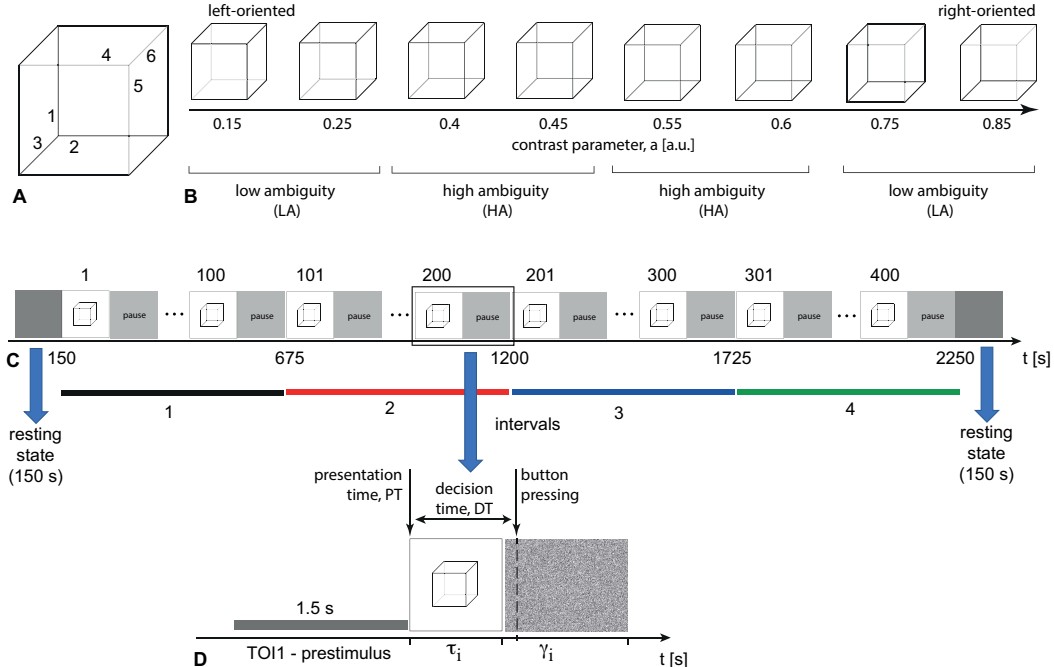

**Figure 1.** Experimental paradigm: (**A**) An example of the Necker cube image with the labeled inner edges. (**B**) Visual stimuli (Necker cubes) with different values of the contrast parameter *a*, which determines orientation and ambiguity. (**C**) Experimental protocol including 150-s resting state recordings and presentation of 400 stimuli alternating with the pauses. Colored horizontal stripes indicate 575-s time intervals. These intervals equally divide the stimuli presentation session. Each interval includes 100 stimuli. (**D**) Detailed illustration of a single stimulus presentation and abstract image. The cube presentation starts at the presentation time PT and lasts $\tau_i \in [1, 1.5]$ s. The decision time (DT) is determined by the interval between PT and button pressing. The pause time $\gamma_i$ varies from 3 to 5 s. The time interval of interest (TOI1) is the 1.5-s pre-stimulus segment time-locked to the PT.

## 2.3. Experimental Protocol

Necker cubes ($22.55 \times 22.55$ cm) were shown on a white background using a 24″ monitor ($52.1 \times 29.3$ cm) with a $1920 \times 1080$ pixels resolution and a 60 Hz refresh rate. The distance between the participant and the monitor was 0.79—0.8 m, and the visual angle was ~0.39 rad.

The duration of the entire experiment was about 40 min for each participant, and included EEG recordings of the eyes-open resting state ($\approx$150 s) before and after the main part of the experiment. Cubes with predefined *a* values (selected from the set in Figure 1B) were randomly presented 400 times during the experimental session. Each cube with a particular ambiguity appeared about 50 times.

Each *i*-th stimulus presentation lasted for time interval $\tau_i$, which ranged from $\tau_{min} = 1$ s to $\tau_{max} = 1.5$ s. The pauses between subsequent presentations of the Necker cube images, $\gamma_i$, ranged from $\gamma_{min} = 3$ s to $\gamma_{max} = 5$ s (Figure 1D) and contained an abstract image demo. The abstract image was the white noise picture (Figure 1D).

## 2.4. Behavioral Estimates

The participants were instructed to press either the left or right key in response to the left or right stimulus orientation, respectively. For each stimulus, we registered presentation time (PT)—the time between the beginning of the experiment and the moment when the current stimulus appeared on the screen. The behavioral response to each stimulus was assessed by measuring the decision time (DT), which corresponded to the time passed from the stimulus presentation to the button pressing (Figure 1C). We also monitored the correctness using error rate (ER) by comparing the actual stimulus orientation with the

subject's response. The actual orientation of the Necker cube was defined by the contrast of the inner edges. Thus, $a = \{0.15, 0.25, 0.4, 0.45\}$ defined the left-oriented cubes, while $a = \{0.55, 0.6, 0.75, 0.85\}$ stood for the right-oriented ones. To define the correctness, we checked whether the subject pressed the left button for $a = \{0.15, 0.25, 0.4, 0.45\}$, or the right button for $a = \{0.55, 0.6, 0.75, 0.85\}$. Otherwise, their response was considered as incorrect. We excluded two subjects with $ER > 20\%$, as they exceeded the 90th percentile of ER distribution in the group.

### 2.5. EEG Recording

For registration of EEG signals, a monopolar method and a classical extended 10–10 electrode scheme were used. We recorded signals from 31 channels using an electrode cap, with two reference electrodes on the earlobes ($A1$ and $A2$) and a ground electrode $N$ above the forehead. Ag/AgCl cup adhesive electrodes placed on the "Tien–20" paste (Weaver and Company, Aurora, CO, USA) were used for signal acquisition. Immediately before the experiments, a special abrasive "NuPrep" gel (Weaver and Company, Aurora CO, USA) was applied to the electrode attachment areas to increase skin conductivity. We maintained the impedance values in the range of 2–5 kΩ. For registration, amplification, and analog-to-digital conversion of the EEG signals, we used a multichannel electroencephalograph "Encephalan-EEG-19/26" (Medicom MTD company, Taganrog, Russian Federation) with a two-button input device (keypad). This device holds the registration certificate from the Federal Service for Supervision in Health Care No. FCP 2007/00124 of 07.11.2014 and European Certificate CE 538571 from the British Standards Institute (BSI).

The raw EEG signals were filtered by a fourth-order Butterworth (1–100)-Hz band-pass filter and a 50-Hz notch filter with built-in acquisition hardware and software. In addition, we performed an independent component analysis (ICA) to remove eye blinking and heartbeat artifacts. To determine components with artifacts, we examined their scalp map projections, waveforms, and spectra. The components containing eye-blinking artifacts usually had leading positions in the component array due to high amplitude. They demonstrated a smoothly decreasing spectrum, and their scalp map showed a strong far-frontal projection. Finally, eye-blinking artifacts had the typical waveform; therefore, those segments of EEG signals were marked by the experienced neurophysiologist, and used for determining the corresponding independent components.

We then segmented the EEG signals into 4-s trials, where each trial was associated with a single presentation of the Necker cube, and included a 2-s interval before and 2-s interval after the moment of the stimulus demonstration. After the EEG pre-processing procedure, we excluded some trials due to the remaining large-amplitude artifacts. To exclude trials containing large amplitude artifacts, we used the $z$-value threshold $z < 1$. The rejection procedure was performed using FieldTrip toolbox in Matlab [21].

After all preprocessing procedures, we had $52 \pm 11$ SD trials for the interval 1, $47 \pm 11$ SD trials for the interval 2, $47 \pm 11$ SD trials for the interval 3, and $55 \pm 11$ SD trials for the interval 4. We calculated the wavelet power for each trial in the (4–40)-Hz frequency range using the Morlet wavelet, and the number of cycles $n$ was defined as $n = f$, where $f$ is the signal frequency. Finally, we computed the event-related spectral perturbation (ERSP) by normalizing the wavelet power estimates $W$ to the wavelet power of 40-s resting-state EEG as $ERSP = (W - W_{rest})/W_{rest}$. All processing procedures were performed offline.

Our goal was to study how the participant's state changed in the course of the experiment, regardless of the type of stimulus. Therefore, we measured brain activity before the start of the stimulus presentation (1.5-s prestimulus interval, TOI1 in Figure 1D).

### 2.6. Source Localization

We applied low-resolution precision electromagnetic brain tomography (eLORETA) to solve the inverse problem and localize the sources of neuronal activity according to EEG data at each of the predetermined points (voxels) in the brain volume [22–24].

LORETA is low-resolution brain electromagnetic tomography. This method solves the inverse problem: converting EEG measurements into information about the distribution of neural sources power into a brain volume. This method belongs to the class of nonparametric methods [25], which are based on the assumption that a separate current dipole (a source) is assigned to each of tens of thousands of elements of the tessellation of the cerebral cortex, while the orientation of the dipole is determined by the local normal to the surface. In this case, the inverse problem is linear, since the only unknowns are the amplitudes of the dipoles. Exact low-resolution brain electromagnetic tomography (eLORETA) is 3D, regularized, and minimum norm-weighted inverse solution with theoretically accurate zero error localization, even in the presence of structured biological or measurement noise [22,25]. The "Colin27" brain MRI averaged template [26] was used to develop a three-layer (brain, skull, and scalp) head model based on a boundary element method (BEM) [27,28]. The sources space inside the brain consisted of 11,865 voxels. The location of the EEG electrodes corresponded to the template head shape.

We analyzed the source characteristics in the predefined time-frequency domain of interest, selected on the basis of sensor-level analysis. To do this, we reassigned the EEG signals to the total average, subtracted the mean, and filtered with a fourth-order Butterworth $[f_L, f_H]$-Hz band-pass filter, where $f_L$ and $f_H$ define the frequency domain of interest. Then, we performed time-lock averaging across the TOI1 trials and computed the covariance matrix. The inverse solution yielded estimates of the source power in each voxel, averaged over the selected TOI window. Finally, we normalized the obtained estimates of the power $P$ of each source to the power of 40-s EEG resting state EEG as $(P - P_{rest})/P_{rest}$. We used the automated anatomical labeling (AAL) brain atlas [29] to map the location of sources to the anatomical brain regions.

### 2.7. Experimental Conditions

Since the observer's state, as a rule, is not at a constant level, but fluctuates at different time scales, in the short term, this causes a difference in the subject's behavior when presenting stimuli, even if their ambiguity does not change. Meanwhile, in the long term, cognitive fatigue occurs, and the training effect takes place. In this work, we have eliminated short-term fluctuations and focused on the long-term changes in the person's condition over a 40-min task. To exclude the influence of short-scale fluctuations, we divided the entire experiment into four consecutive intervals of 10-min duration each (see Figure 1B). For each interval, we averaged ERSP and source power (SP) over all trials. To assess the behavioral characteristics, we used the median DT and the error rate (ER), reflecting the percentage of erroneous responses at each interval.

### 2.8. Statistical Testing

We tested how DT and ER changed at 1–4 intervals using additional controls of stimulus orientation and ambiguity. We performed repeated measures ANOVA with 1–4 intervals, ambiguity (HA and LA), and orientation (Left and Right) as within-subject factors. In general, ANOVA requires the homogeneity assumption: the population variances of the dependent variable must be equal for all groups. At the same time, this assumption may be ignored if the sample size is equal for each group. In this study, we used a within-subject design with repeated measures. Therefore, the sample size for each condition was equal, and we did not control for variance homogeneity. If the tested samples did not obey the normality condition, we applied Greenhouse–Geisser correction to ANOVA results. For significant main effects, we performed a post hoc analysis using parametric or nonparametric tests, depending on sample normality, which was determined using the Shapiro–Wilk test. All test types are specified in the Result section and in the figures captions. A statistical analysis was performed in IBM SPSS Statistics.

Statistical analyses of brain activity were carried out based on the subject-level wavelet power, averaged over trials and over TOI1. Contrasts between the four intervals were tested for statistical significance using a permutation test combined with the cluster-based

correction for multiple comparisons. Specifically, the *F*-tests compared four wavelet power sets for all pairs (channel, frequency). Items that passed the threshold corresponding to a *p*-value of 0.001 (one-tailed) were labeled along with their adjacent items and collected in separate negative and positive clusters. The minimum required number of neighbors was set to 2. The *F*-values in each cluster were summarized and corrected. The maximum amount was entered into the permutation structure as a test statistic. A cluster was considered significant if its *p*-value was below 0.01. The number of permutations was 2000.

A similar procedure was followed on the source level results. We performed a cluster-corrected statistical intra-subject permutation test on the test-averaged and TOI1-averaged source power distributions to determine significant differences between four intervals [30,31]. The threshold for paired comparisons with *F*-test was $p = 0.005$. The p-threshold for the cluster was 0.025. The number of permutations was 2000. Finally, we calculated, for each subject, the average power of the source activity in the region of the identified cluster for each of the four intervals.

All described operations were performed in Matlab using the Fieldtrip toolbox [21,32].

## 3. Results

### 3.1. Results of the Behavioral Data Analysis

Contrasting subjects' DTs on four intervals, we observed significant changes among intervals, the effect of stimulus ambiguity, and the combined effect of ambiguity and orientation (see Table 1). Nevertheless, the experiment demonstrated that DT varies over the course of the experiment in the same way for all stimuli. We also found a significant effect of ambiguity on ER. At the same time, we observed a significant interaction effect of interval and ambiguity (see Table 2). These results suggest that ER differs for HA and LA stimuli regardless of their orientation. Moreover, ER changed differently during the experiment depending on the ambiguity.

**Table 1.** Median decision time to the current stimulus, DT [s] (ANOVA Summary).

| Factors | $dF_1$ | $dF_2$ | $F$ | $p$ |
|---|---|---|---|---|
| Interval | 2.285 | 38.853 | 5.805 | 0.005 * |
| Ambiguity | 1 | 17 | 79.524 | <0.0001 * |
| Orientation | 1 | 17 | 1.093 | 0.310 |
| Interval × Ambiguity | 3 | 51 | 1.206 | 0.317 |
| Interval × Orientation | 3 | 51 | 1.290 | 0.288 |
| Ambiguity × Orientation | 1 | 17 | 6.385 | 0.022 * |
| Interval × Ambiguity × Orientation | 3 | 51 | 0.544 | 0.655 |

Here, '*' indicates the level of significance $p < 0.05$.

**Table 2.** Percentage of erroneous responses to the current stimulus, ER [%] (ANOVA Summary).

| Factors | $dF_1$ | $dF_2$ | $F$ | $p$ |
|---|---|---|---|---|
| Interval | 1.995 | 33.910 | 2.988 | 0.064 |
| Ambiguity | 1 | 17 | 13.128 | 0.002 * |
| Orientation | 1 | 17 | 2.671 | 0.121 |
| Interval × Ambiguity | 3 | 51 | 5.918 | 0.002 * |
| Interval × Orientation | 1.975 | 33.58 | 1.454 | 0.248 |
| Ambiguity × Orientation | 1 | 17 | 0.922 | 0.350 |
| Interval × Ambiguity × Orientation | 1.63 | 27.715 | 1.892 | 0.175 |

Here, '*' indicates the level of significance $p < 0.05$.

The post hoc analysis revealed that subjects responded faster to LA stimuli than HA ones: $Z = -3.724, p < 0.0001$, Wilcoxon test (Figure 2A). For HA stimuli, DT was similar for the left- and right-oriented stimuli: $t(17) = 0.383, p = 0.706$, *t*-test (Figure 2B). For LA stimuli, subjects responded faster to the left-oriented stimuli: $Z = -2.591, p = 0.01$, Wilcoxon test (Figure 2C). Analysis of pairwise differences displayed that 12 subjects demonstrated effects in the same direction as the group. Finally, DT decreased with the

interval: $\chi^2(3) = 12.218$, $p = 0.007$, Friedman test (Figure 2D). The post hoc Wilcoxon test revealed higher ER for HA stimuli when compared to LA stimuli: $Z = -3.29$, $p = 0.001$ (Figure 2E). For HA stimuli, ER decreased during the course of the experiment: $\chi^2(3) = 7.545$, $p = 0.056$, Friedman test (Figure 2F). Finally, there was no correlation between age and DT to HA stimuli: $r(20) = -0.24$, $p = 0.3$ and LA stimuli: $r(20) = -0.31$, $p = 0.17$. DT was similar for males and females for both HA stimuli: $t(18) = 0.79$, $p = 0.436$ and LA stimuli: $t(18) = 0.96$, $p = 0.348$.

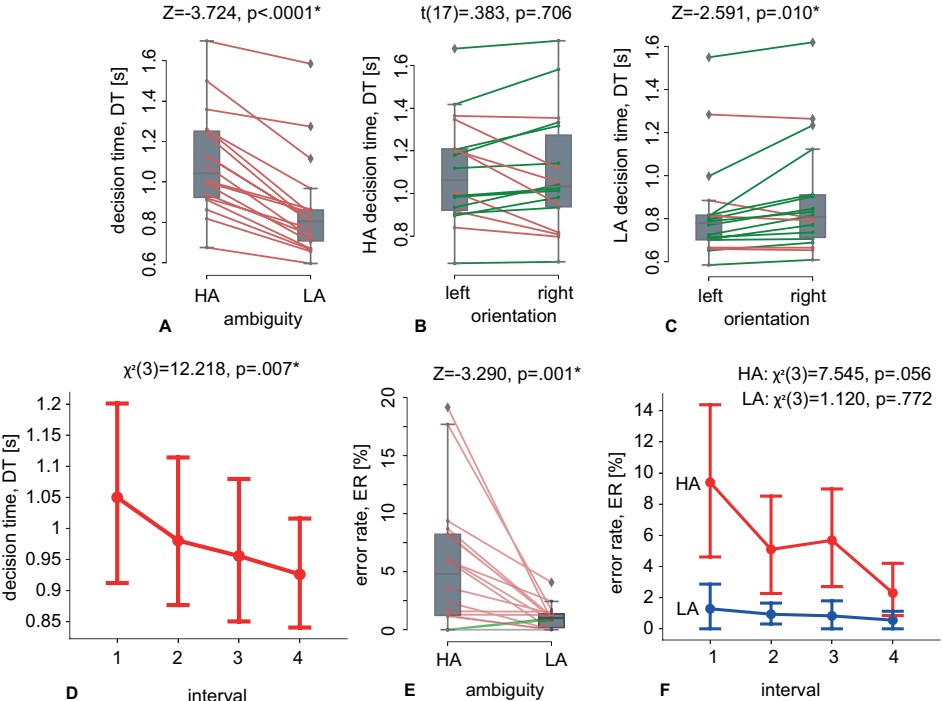

**Figure 2.** Results of the behavioral data analysis: (**A**) Median decision time, DT to HA and LA stimuli (*$p < 0.0001$ via Wilcoxon test, uncorrected). (**B**) Median DT to the left- and right-oriented HA stimuli ($p = 0.706$ via *t*-test, uncorrected). (**C**) Median DT to the left- and right-oriented LA stimuli (*$p = 0.01$ via Wilcoxon test, uncorrected). (**D**) Median DT (group mean $\pm$ 95% CI) on four intervals (*$p = 0.007$ via Friedman test, uncorrected). (**E**) Percentage of erroneous responses (ER) to the HA and LA stimuli (*$p = 0.001$ via Wilcoxon test, uncorrected). (**F**) ER (group means $\pm$ 95% CI) on all intervals separately for HA ($p = 0.056$ via Friedman test, uncorrected) and LA ($p = 0.772$ via Friedman test, uncorrected) stimuli.

### 3.2. Results of the EEG Data Analysis on the Sensor Level

Comparing the prestimulus ERSP on four intervals, we found one cluster with $p = 0.0015$ (corrected using the permutation statistics) in the 9–11 Hz frequency band, including parietal and temporal sensors (P8, TP8, T8, FC4, FT8) in the right hemisphere (Figure 3A). The ERSP averaged over the EEG sensors in this cluster grew with the interval number from $0.18 \pm 0.11$ SE at interval 1 to $0.49 \pm 0.15$ SE at interval 4 (Figure 3B). We also compared the ERSP between males and females and between two age groups ("<26 y.o." vs. ">26 y.o.", where 26 was the median age) via the independent-samples *t*-test with cluster-based correction for multiple comparisons. In both cases, the difference was insignificant.

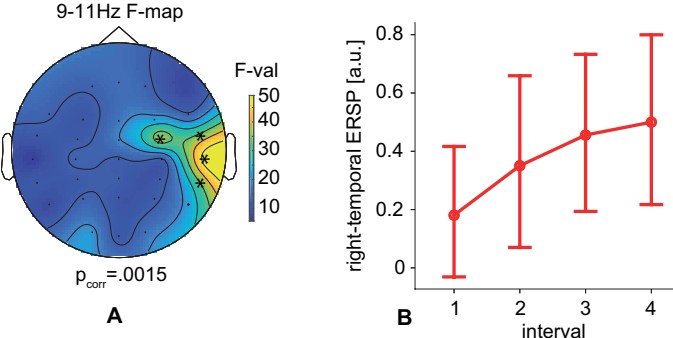

**Figure 3.** Results of the EEG data analysis on the sensor level: A scalp topogram illustrates *F*-value and EEG channels cluster, demonstrating the significant change of ERSP between four intervals (\* $p = 0.0015$ via *F*-test and cluster-based correction for multiple comparisons) (**A**). Changing ERSP in this cluster (group mean $\pm$ 95% CI) with the time with the interval number (**B**).

### 3.3. Results of the EEG Data Analysis in the Source Space

Setting the bandwidth of interest to $10 \pm 2$ Hz, we contrasted the source power (SP) between the four intervals. The statistical analysis revealed one cluster in the source space with $p = 0.02$ (corrected using the permutation statistics). This cluster included voxels in the right middle temporal gyrus (Temporal Mid R), right superior temporal gyrus (Temporal Sup R), right inferior temporal gyrus (Temporal Inf R), Rolandic operculum (Rolandic Oper R), fusiform gyrus (Fusiform R), and a part of the cerebellum (Cerebellum Crus1 R) (Figure 4A). The maximal *F*-value was achieved in the right middle temporal gyrus, while the minimum *F*-value belonged to the cerebellum. The averaged SP in this cluster grew from interval 1 ($-0.22 \pm 0.12$ SD) to interval 4 ($1.21 \pm 0.49$ SD) (Figure 4B).

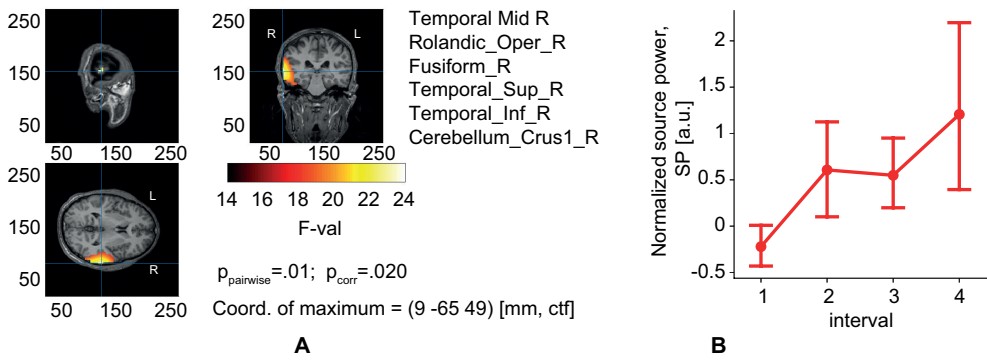

**Figure 4.** Results of the EEG data analysis in the source space: (**A**) Source plot shows *F*-value, reflecting the significant change of the source power (SP) between four intervals on the prestimulus interval $t \in [-1.5, 0]$ s, for $f \in 8 - 12$ Hz ($p = 0.02$ via *F*-test, permutation-based correction). Legends contain *p*-values, CTF coordinates of the voxel with maximal *F*-value, and names of anatomical zones according to Automated Anatomical Labeling (AAL); (**B**) Normalized source power, NSP (group mean $\pm$ 95% CI) in this cluster on four intervals.

### 3.4. Results of the Correlation Analysis

Using repeated measures correlation analysis, we found that SP was negatively correlated with DT to LA stimuli (Figure 5A) and ER for HA stimuli (Figure 5D). At the same time, there was no correlation between SP and DT to HA stimuli (Figure 5B), or between SP and ER for LA stimuli (Figure 5C).

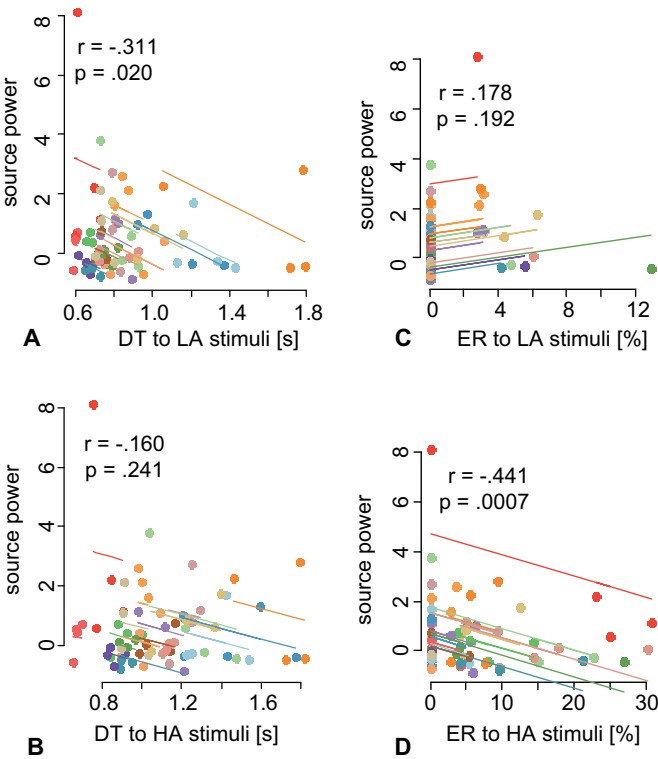

**Figure 5.** Results of the correlation analysis: Regression plots illustrate the relationship between SP and DT to LA stimuli (**A**); SP and DT to HA stimuli (**B**), SP and ER for LA stimuli (**C**); SP and ER for Ha stimuli (**D**). The colored dots correspond to each participant's data; the lines have the same slope estimated for these participants via correlation analysis with repeated measures.

## 4. Discussion

A group of volunteers was tasked to classify Necker cubes of different ambiguity within 40 min. The subjects reported the orientation (left or right) of each presented cube, while the stimulus morphology ranged from low ambiguity (LA) to high ambiguity (HA). By observing behavioral responses, we found that decision time decreased with the time on task for both HA and LA stimuli (Figure 2D). At the same time, the subjects improved the correctness of the interpretation of the HA stimuli, but not the LA stimuli (Figure 2F). Analysis of the EEG spectral power on the sensor level and in the source space revealed an increase in the prestimulus power at 9–11 Hz with the time on the task (Figures 3B and 4B). Finally, we found that the prestimulus EEG power negatively correlated with the decision time to LA stimuli (Figure 5A) and the number of erroneous responses to HA (Figure 5D) stimuli.

First, we hypothesized that the prestimulus EEG power reflects changes in a person's condition. The condition, in turn, affected the performance of processing the ongoing visual stimulus [11,16]. Thus, our results showed that the high pre-stimulus 9–11 Hz EEG power predicted faster decision times and greater accuracy. It is worth noting that we compared EEG power and behavioral estimates between time segments, each of which lasted 10 min. Therefore, we have associated the described effects with slow changes in the observer's state. Taken together, we proposed a possible application of our findings in passive brain-computer interfaces to monitor the human's condition and predict decision speed and errors.

Along with possible practical applications, our result can help to reveal specific features of ambiguous perception. To start a discussion on this aspect, we will look at the limitations of our experimental design. It consists of two confounding variables, mental fatigue, and learning; both can be time-dependent. Thus, the revealed EEG changes practically do not reflect one of these factors. At the same time, we suppose that fatigue

and learning affect the observer state regardless of the stimulus ambiguity. In contrast, our results show a different time-on-task effect on processing HA and LA stimuli. For instance, we revealed a reduction in error rate for HA stimuli only. In addition, the prestimulus EEG power negatively correlates with the decision time to LA stimuli, not to HA stimuli, and negatively correlates with the error rate for HA, but not for LA stimuli. Another limitation is the small sample size. Thus, there is a risk that the individual characteristics of the participants, e.g., gender and age, will affect their perception of ambiguous stimuli. For our group, we observed no gender and age effects on the decision time and the ERSP, due to the almost uniform distribution of these factors between participants. At the same time, we expect that another group of younger or older subjects may demonstrate different effects on the behavioral and neural activity levels.

The decision time (DT) may decrease due to neural adaptation (NA), which occurs when the same visual stimulus is repeatedly presented within a short interval and causes a decrease in neural response to repetitive versus non-repeated stimulus [33]. The NA is thought to arise from at least two types of neural activity. One explanation is that only the part belonging to the neuronal ensemble is sensitive to stimulus recognition. Thus, the neurons that are not critical for stimulus recognition decrease their responses when the stimulus reappears, while on the contrary, neuronal populations carrying essential information continue to give a robust response. As a result, the mean firing rate decreases due to stimulus repetition [34]. An alternative explanation is that stimulus repetition reduces response in the time domain [33]. According to this theory, a neural network that processes sensory information responds faster to a repetitive stimulus than to a new stimulus, i.e., a stable response. The network connections involved in the response creation were reinforced by the previous presentation of the same stimulus [35]. The NA affects the neuronal response in the occipital [36], parietal [37], and frontal [38] cortex areas in single-unit data, and on the sensory level. As known from the literature, NA, as a rule, reduces the stimulus-related EEG/ECoG response to stimuli. Here, we did not consider the post-stimulus EEG power and did not report such signs of NA. At the same time, an increase in the prestimulus EEG power may reflect the preactivation of sensory neurons. We suppose that in this preparatory state the neural ensemble exhibits less activation in response to the stimulus. Further research should verify this hypothesis by examining the post-stimulus activity as a function of the time-on-task.

Another potential explanation is the vital role of alpha-band oscillations for visual perception. Our results showed that an increase in the 9–11-Hz band power correlates with enhancing processing performance. In contrast, many studies have reported negative effects of alpha power on processing performance. For example, the authors of Ref. [12] reported that moderate to low alpha signal strength in the prestimulus period is beneficial for sensory perception. In contrast, a recent review [39] emphasizes that high alpha power facilitates perception by suppressing irrelevant input and generating predictors in the visual cortex. The latter was confirmed by observing an increase in the prestimulus alpha power when participants could predict the identity of the forthcoming stimulus [40].

Finally, the role of alpha-band oscillations depends largely on their incident brain region. For instance, the authors of Ref. [41] provided evidence that right temporal alpha oscillations play a crucial role in inhibiting habitual thinking modes, thereby developing creative cognition. Another work showed that observing a Necker cube can improve subsequent creative problem-solving [42]. In line with these works, we supposed that increasing 9–11 Hz power in the right temporal region reflects a developing ability to inhibit obvious associations. According to Ref. [42], the latter may be a biomarker of neural processes facilitating creative problem-solving.

We hypothesize that NA affects the bottom-up processing, while the other two represent the top-down processing components [43,44]. Assuming that NA facilitates the bottom-up processing, we provide a possible explanation for the negative correlation between the prestimulus EEG power and the decision time to LA stimuli. The morphology of inner edges unambiguously determines the orientation of the LA stimulus. Consequently,

during the LA stimulus processing, the bottom-up component prevails [45]. If so, neurons in sensory areas receive the information needed to make the right decision. Repeated stimulation pre-activates these sensory neurons, which leads to a decrease in decision time to LA stimuli. In contrast, the HA stimuli processing may require top-down mechanisms, because the morphology remains similar for the cube being either left or right-oriented [20]. To explain the lack of correlation between the prestimulus EEG power and decision time to HA stimuli, we also assume that during HA processing, the top-down component predominates for most of the time interval, while the bottom-up one may be limited to an earlier and shorter time window [16]. Thus, by facilitating the bottom-up processing, NA has little or no effect on the overall decision time for HA stimuli.

Summing up, we can say that the increasing prestimulus EEG power can reflect NA in sensory neural networks encoding the Necker cube morphology, which explains the negative correlation between the EEG power and the decision time to LA stimuli. At the same time, decision time to HA stimuli also decreases with time on task, but barely correlates with the prestimulus EEG power. This behavioral effect is probably the result of neural processes acting after the stimulus onset and relies on the integrative dynamics, rather than the EEG power modulation in a particular area. In our future studies, we will address this issue by considering the functional connectivity evolution during the experiment.

By examining the error rate, we found that the observer responded more correctly to LA stimuli. The number of erroneous responses is less than 2%, and remains unchanged during the experiment. In contrast, the number of incorrect responses to HA stimuli decreased with time on task and negatively correlated with the prestimulus 9–11 Hz EEG power. This effect may be a result of the top-down mechanisms, e.g., predicting the identity of the forthcoming stimulus or creative thinking. As discussed above, these processes also accompany increasing alpha-band power and relate to ambiguous stimuli processing.

## 5. Conclusions

During the Necker cube classification, participants decrease their decision time with the time on task, regardless of the stimulus ambiguity. At the same time, they improve the correctness of interpretation only for the highly ambiguous stimuli. EEG analysis has revealed growing prestimulus alpha-band power with the time on task. We have found that the prestimulus EEG power negatively correlates with the decision time for the stimuli, with low ambiguity and the number of erroneous responses to highly ambiguous stimuli.

We suppose that repetitive stimuli presentation affects top-down and bottom-up processing mechanisms. Thus, it may cause the neuronal adaptation of the sensory neurons facilitating bottom-up processing. For the low ambiguity, the bottom-up component dominates; therefore, decision time correlates with the prestimulus EEG power. Increasing alpha-band EEG power may also reflect modulation of top-down components, e.g., the ability to suppress irrelevant information and form the predictors of ongoing stimulus. For the high ambiguity, the top-down processes dominate; therefore, the correctness rate correlates with the prestimulus EEG power.

Finally, our results may add to uncovering the associations between the current human condition and their performance in solving ongoing tasks. It is essential for BCIs to aim not only at the detection, but also the prediction of the human states. These BCIs will give rise to the artificial intelligence systems that assist or alarm when detecting a high probability of critical errors.

**Author Contributions:** Conceptualization, A.N.P.; Data curation, A.E.H.; Formal analysis, S.A.K. and A.N.P.; Funding acquisition, A.E.H.; Investigation, A.K.K. and S.A.K.; Methodology, V.A.M. and A.E.H.; Project administration, A.E.H.; Software, A.K.K.; Validation, V.A.M.; Visualization, S.A.K. and V.A.M.; Writing—original draft, V.A.M.; Writing—review and editing, A.N.P. and A.E.H. All authors have read and agreed to the published version of the manuscript.

**Funding:** This research was funded by the Russian Science Foundation (Project 19-72-10121) V.A.M. supported by the Russian Foundation for Basic Research (19-32-60042) in part of the experimental paradigm development and behavioral data analysis. S.A.K. is supported by the President Program (MD-1921.2020.9) in the part of source reconstruction.

**Institutional Review Board Statement:** The study was conducted according to the guidelines of the Declaration of Helsinki, and approved by the Institutional research ethics committee of the Innopolis University (protocol 2 from 18 January 2019).

**Informed Consent Statement:** Informed consent was obtained from all subjects involved in the study.

**Acknowledgments:** The authors graciously acknowledge the subjects for their contribution to this research.

**Conflicts of Interest:** The authors declare no conflict of interest. The funders had no role in the design of the study; in the collection, analyses, or interpretation of data; in the writing of the manuscript, or in the decision to publish the results.

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
