# Peer review of "Monitoring Brain State and Behavioral Performance during Repetitive Visual Stimulation"

_applsci, doi:10.3390/app112311544_

Round 1

Reviewer 1 Report

Ambigous perception is a crucial feature in visual system interpretation of two or more distinct forms of graphical images like the Neker cubes. The Authors used the Necker cubes of different level of ambiguity and different orientation to study changes in prestimulus neural activity to predict perceptual performance on behavioural level. The manuscript presented for the review provides information on the correlations between prestimulus EEG power to response time and error rate, which depends on difficulty level of the task. The authors indicate that it is possible to predict perceptual performance by monitoring prestimulus EEG power.

The research topic undertaken by the authors is interesting and purposeful. The introduction briefly highlight the study assumptions. Materials and methods are described in great detail and the study is correctly designed. The Authors provide an interesting and accurate description of the results.

As the reviewer, I have several comments about the manuscript:

  • The number of the study participants is quite small, so the results may be misinterpreted in relation to individual cognitive abilities and perception of ambiguity
  • It should be explained in one of the manuscript sections why there was no division within participants into male and female group. Why the sex dimorphism in brain structure and cognitive abilities were not considered.
  • Abbreviations in line 72 should be better explained. Is the bl for the “left brightness” and br for the right brightness? The abbreviation “a” is also not explained.
  • Lines 79-82. These assumptions should be placed in discussion section while there are also references placed in the end of the sentence.
  • Line 158. The abbreviation “PT” should be explained in the text in the place of its first appearance
  • Lines 158-153. In the section of Materials and methods, the authors mention that they performed repeated ANOVA tests with different post hoc tests depending on the results of Shapiro-Wilk normality test. Firstly, ANOVA is also sensitive for variance homogeneity, not only for normality condition. Were there any variance homogeneity tests before ANOVA? Secondly, if there were results that did not fulfil the normality condition, there should be a nonparametric test used instead of ANOVA. Maybe the paragraph should be retyped to be more precise and understandable.
  • In Table 1 and 2, there is no explanation of the number of asterixes. What does it mean if there are two, three or four astericses. I presume that it means the significance of the difference, but it is well visible by analysing the p value.
  • Lines 208-212 and 252-257. The paragraphs are repeated. Please correct the discussion removing repetitions.
  • Line 238-242. “As a result, the mean firing rate decreases due to stimulus repetition.” Are there any studies of other authors to confirm the thesis?
  • Line 246. “As known from the literature…” please point this literature in references.
  • Line 267-280. Lots of hypotheses without support in literature. Please discuss this paragraph with other authors’ articles.

In my opinion, the manuscript after minor revision will be suitable for publication in Applied Sciences.

Reviewer 2 Report

Kuc and colleagues present here a study trying to evaluate the neural correlates of behavioral performance depending on repetitive stimuli. The manuscript is well-written, but I am not sure regarding the results presented. Therefore, the authors’ conclusions do not seem entirely accurate. Please see my detailed concerns below. 

Introduction

  • It is missing the introduction of previous evidence regarding the neuronal signatures of different mental states. 
  • Authors should better differentiate this current study from their own previous ones using this experimental paradigm. 
  • The gap from the current research question and its application to BCI is quite high. Please consider improving the proposed bridge between both.

Methods

  • ‘’They were familiar with the experimental task and had not participated in similar experiments for the past six months’’. Which kind of familiarity did the participants have with the task?
  • Please provide the stimulus dimensions in visual angles. 
  • The randomization processes are basic procedures without the need for such detailed descriptions.
  • What was the abstract image used during pauses? 
  • Was there any pause during the stimulation session or were the data acquired continuously during the 40 minutes? Did the authors control for fatigue effects?
  • Response time might be interpreted as reaction time. I suggest renaming this interval. Or does the interval only last for the reaction time calculated? 
  • Did the authors use an electrodes cap or were they directly placed in the participants’ heads?
  • It is not clear whether the EEG data processing steps described in the EEG recordings section were performed online or offline. 
  • Did the authors re-referenced the EEG data? 
  • Was some EOG channel acquired for control processes?
  • How did the authors combine both reference electrodes? 
  • How did the authors choose which ICA components to remove? And the trials to remove due to large-amplitude artifacts? What were the criteria? 
  • How many trials were analysed per condition? 
  • Please provide the exact eLORETA meaning. 
  • How did the authors define correct vs error for conditions? This is particularly relevant for conditions with high levels of ambiguity. 

Results

  • Based on figure 3, the alpha pattern reported seems to be focused in the right temporal region. Is that right? 
  • I would like to see the ERSP results. 
  • What was the baseline used for the analysis? 
  • I would like to see the source analysis results. 
  • The behavioral results should also be presented. 

Discussion

  • It is several times used the term ‘’the prestimulus EEG power’’. What do the authors mean? 
  • The alpha oscillations neural correlates depend largely on their incident brain region. It should be taken into account to validate your results and conclusions. The discussion provided is vague and incomplete. 
  • ‘’We suppose that these differences reflect the essential features of ambiguous processing.’’ What would be the essential features of ambiguous processing? 
  • Was the erroneous and correct data unbalance taken into account?

Reviewer 3 Report

Strong aspects:
This research evaluates the correlation between the intensity of the pre-stimulus brain activity and its performance in solving tasks. These results are of high importance for the design of active brain-computer-interfaces.

Weak aspects:
The discussions section should be revised in order to improve the correlation between the presented results in the figures and the statements.

Comments to the authors:
- The lines 11-15 in the abstract should present more clearly how LA and HA stimuli correlates to EEG power. 
- It is not clear what PT means (line 158)
- It is not clear that LO and RO means left oriented and right oriented.(line 160)
- References to the results presented in Figs. 2 and 3 should be added with the statements:
- "our results showed that the high pre-stimulus EEG power predicted faster response times and greater accuracy." (lines 216, 217)
- "Analysis of the EEG spectral power revealed an increase in the prestimulus power  at 9–11 Hz with the time on the task." (lines 211-212)
- The last two paragraphs in the discussions section should be included in the Conclusions section which is missing.

Typos:
The word 'intstructed' should be revised.
